# Your Context Is Not an Array:
# Unveiling Random Access Limitations in Transformers

**M.Reza Ebrahimi**
University of Toronto, Qualcomm AI Research*
ebrahimi@qti.qualcomm.com

**Sunny Panchal**
Qualcomm AI Research
sunnpanc@qti.qualcomm.com

**Roland Memisevic**
Qualcomm AI Research
rmemisev@qti.qualcomm.com

## Abstract

Despite their recent successes, Transformer-based large language models show surprising failure modes. A well-known example of such failure modes is their inability to length-generalize: solving problem instances at inference time that are longer than those seen during training. In this work, we further explore the root cause of this failure by performing a detailed analysis of model behaviors on the simple parity task. Our analysis suggests that length generalization failures are intricately related to a model's inability to perform random memory accesses within its context window. We present supporting evidence for this hypothesis by demonstrating the effectiveness of methodologies that circumvent the need for indexing or that enable random token access indirectly, through content-based addressing. We further show where and how the failure to perform random memory access manifests through attention map visualizations.

## 1 Introduction

The evolution of Transformer-based large language models (LLMs) has marked a new era in how machines understand and interact with human language. Their capabilities extend far beyond natural language tasks, encompassing instruction following (Ouyang et al., 2022), code generation (Zhang et al., 2023), theorem proving (Wu et al., 2022), and common sense and multi-step reasoning (Yu et al., 2023). This has made LLMs play a pivotal role as the backbone of AI agents (Xi et al., 2023), and even has sparked discussions around their ability to exhibit glimpses of general intelligence (Bubeck et al., 2023).

Despite these remarkable capabilities, surprisingly, the same models struggle with seemingly simple arithmetic tasks, such as multi-digit addition and multiplication (Dziri et al., 2024). Specifically, the models fail to learn simple algorithms to perform these arithmetic operations. This becomes apparent when models are applied to problems of greater length than those encountered during training (Hupkes et al., 2020), a problem setting generally referred to as *length generalization*.

Arithmetic tasks fundamentally differ from natural language tasks in two key aspects. First, unlike natural language, responses to arithmetic tasks are objective and unambiguous, corresponding to the exact execution of a sequence of algorithmic steps. The second difference, and the focus of our work, is their reliance on formatting: arithmetic expressions are represented using a limited vocabulary, such as digits, with each token holding equal significance.

Crucially, in the representation of arithmetic tasks, a token's position is as important as its value. This stands in stark contrast to natural language expressions, in which the coupling

---

*Qualcomm AI Research is an initiative of Qualcomm Technologies, Inc.

between token or word positions on the one hand and the meaning of the expression on the other is much weaker and much more flexible. In the context of language modeling this has been demonstrated, for example, by Sinha et al. (2021), who show that permuting word orders has a surprisingly small effect on the performance of BERT models in natural language processing tasks.

In other words, the meaning of natural language utterances depends largely on the meaning of their constituents (*e.g.*, words) and only partially on their positions. This well-known influence of meaning (semantics) over pure syntax is exemplified in expressions, such as "He saw the cat with the binoculars", in which the phrase "with the binoculars" is more likely subordinate to "He", even though syntactically it could equally be subordinate to "the cat". The precise position of individual words becomes even less informative when references stretch over larger distances, such as across sentences.

As illustrated in Figure 1, when predicting the next token in a natural language task, token references which are *"content-based"* in this way are well represented by the common attention mechanism prevalent in the Transformer, and they are further reinforced through pre-training on natural language. This is in contrast to arithmetic tasks, which rely exclusively on *"index-based addressing"* (random access memory) into the context window to retrieve the information necessary for generating the next algorithmic step.

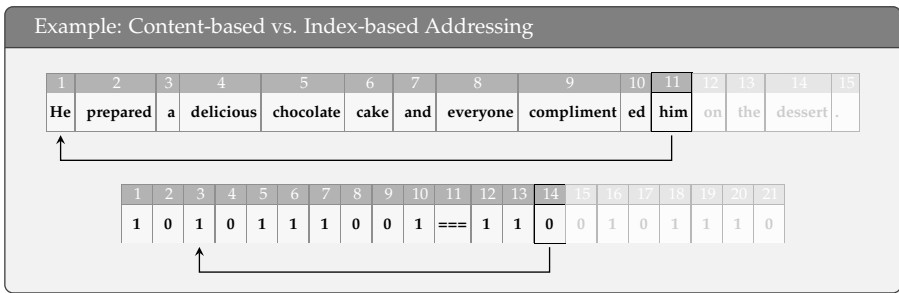

Figure 1: **Top:** Prediction in natural language tasks. To predict the pronoun `him`, the model needs to access previously used pronouns in the context, among other tokens, regardless of the exact position of the token `He` in the context (content-based addressing). **Bottom:** Prediction in an arithmetic task. The model returns the running parity of the binary sequence after ===. For the third output, the model must precisely attend to the token in position 3 of the context window (index-based addressing).

In this work, we provide an in-depth study of this addressing dichotomy and present evidence for its role in the failure of Transformer language models in algorithmic tasks. We focus on the binary parity task as it is, arguably, the simplest sequential arithmetic task, making it well-suited to study the underlying computational requirements of Transformers applied to it. When properly formatted, the state needed to carry over at each step is only one bit, and the key operation required to learn is XOR. Yet, Transformer-based models struggle to learn a length generalizable algorithm for this task (Anil et al., 2022).

Our detailed empirical study of the parity task across models with various positional embedding methods strongly supports the hypothesis that Transformers pre-trained on natural language learn to retrieve tokens using content-based addressing, leading them to fail on algorithmic tasks which, as discussed, depend on random memory access.

In Sections 3 and 4, we further demonstrate how the addition of *"mnemonics"* to leverage content-based addressing as a workaround for index-based addressing allows models to learn length generalizable algorithms for the parity and addition tasks, both of which were previously shown to be hard for Transformer language models. While the introduction of mnemonics is not proposed as a practical fix, it highlights the underlying issue and reinforces our hypothesis. Our work suggests that equipping models with effective index-based addressing mechanisms could be a key to learning algorithms that can length-generalize.

## 2   Related Work

Length generalization is a well-known problem in the context of Transformer-based sequence models (Qian et al., 2022; Newman et al., 2020; Zhang et al., 2022b; Zhou et al., 2024; Xiao & Liu, 2023). Notably, Anil et al. (2022) conducted careful empirical studies exploring the length generalization capabilities of Transformer-based LLMs with a focus on the boolean variable assignment and binary parity task. They demonstrated that models, even when fine-tuned on these tasks using a scratchpad format, struggle significantly with generalization, regardless of a model's scale.

The study by Dziri et al. (2024) examines the ability of Transformers to length-generalize in compositional tasks, such as multi-digit multiplication, and highlights their generalization failures across zero/few-shot and fine-tuning regimes, both with and without the use of a scratchpad. It suggests that Transformers may approach compositional tasks by simplifying multi-step reasoning into a form of linearized subgraph matching, rather than developing systematic problem-solving skills.

The work by Zhou et al. (2022) examines the extent of in-context learning for algorithmic tasks through the strategic use of meticulously designed prompting techniques, called algorithmic prompting. As we shall show, our work suggests an alternative interpretation for the results of that work based on indexing. Similarly, Zhou et al. (2023) build on the RASP computational model proposed by Weiss et al. (2021), and focuses on identifying algorithmic tasks learnable by transformers. It conjectures that Transformers demonstrate strong length generalization for tasks that can be solved by a concise RASP program across various input lengths.

The work presented in Kazemnejad et al. (2024) involves a systematic comparison of length generalization performance across Transformers with various positional encoding schemes. It reveals that none of the commonly used positional embedding methods effectively solve the length generalization problem in downstream tasks. Surprisingly, having no positional embedding outperforms these methods, echoing a finding previously identified by Shen et al. (2023). This observation further indicates that current positional embedding approaches fail to equip the model with the capability for proper index-based addressing. Moreover, Shen et al. (2023) propose a modification to the positional embedding itself, by marking tokens with random tags. This allows the model to distinguish identical tokens appearing in different positions, offering a slight improvement in generalization.

A study similar in spirit to our work is Dubois et al. (2019), albeit using recurrent sequence-to-sequence models instead of Transformers. That work hypothesizes that models equipped with separate content and location-based attention mechanisms are more likely to be able to extrapolate. It evaluates this hypothesis through variants of the Lookup Table task, designed to directly assess a model's performance in index-based addressing.

The work by Mohtashami & Jaggi (2024) proposes a method for handling long contexts by using sparse learnable "landmark tokens" to retrieve relevant token blocks. These landmark tokens bear some similarity with our use of "mnemonics" we shall discuss below.

## 3   Random Accessing in LLMs – A Case Study

In this section, we focus on the binary parity task as a case study on learning algorithmic tasks with Transformers. We chose the parity task for its simplicity as one of the most basic sequential arithmetic tasks. With the correct scratchpad format, it requires carrying over just one bit of state at each step, and the primary operation to learn is XOR. However, it is known that Transformer-based models struggle to learn the correct algorithm as their solution fails for sequences longer or shorter than those seen during training (Anil et al., 2022).

We begin with a brief note on the usage of scratchpads. When the model is asked to directly output the final answer, such as the parity of a sequence, we encounter a potential complication: Transformers execute a fixed amount of computation for each token generated, yet the problem size can vary. In other words, the model must simulate a for-loop over the entire

sequence in a single forward pass. Note that this represents a distinct contaminating issue that falls outside the scope of this work. This challenge can be addressed by incorporating a "scratchpad" (which is also referred to as chain-of-thought) (Nye et al., 2021; Wei et al., 2022). The scratchpad enables the effective use of the context window to explicitly simulate a for-loop and output intermediate results.

Adopting the format used in Anil et al. (2022) for the parity task, we begin with a start-of-sequence symbol >>>, followed by a binary sequence, an end-of-sequence symbol ===, and the sequence's running parity. For instance:

| | |
|---|---|
| **No Scratchpad** | >>> 1 0 1 0 0 1 1 === 0 |
| **Standard Scratchpad** | >>> 1 0 1 0 0 1 1 === 1 1 0 0 0 1 0 |

Throughout the paper, `blue bold` tokens are used to indicate tokens over which the loss is calculated during training, and thus also the tokens that the model predicts during inference. Meanwhile, other tokens are added externally into the model's context during generation (via "environment forcing" (Recchia, 2021)). Also, we ensure that the start-/end-of-sequence symbols are converted to single tokens and bits within the sequence are represented by single fixed tokens, preventing any merging due to tokenization.

## 3.1 Interleaved scratchpad

In essence, a length generalizable solution to generate the running parity in the specified format involves three steps: 1) Reading the current active bit; 2) Reading the current running parity, and; 3) Performing XOR between the active bit and the current parity. We hypothesize that the failure of Transformers can be attributed to the first step, since the subsequent two steps are straightforward: the current running parity is the last token generated, and the XOR operation is trivial to learn.

To support this claim with empirical evidence, we implement an *interleaved* scratchpad format where sequence bits and running parities are alternated, ensuring that at each step, the current active bit is the last token, and the current running parity appears immediately before the last token in the context. This arrangement dramatically simplifies the first step (reading the current active bit), which, as we will see shortly, lets the model learn a length generalizable solution.

| | |
|---|---|
| **Interleaved Scratchpad** | >>> 1 1 0 1 1 0 0 0 0 0 1 1 1 0 |

We fine-tuned several small Transformer models with different positional embedding methods: BLOOMZ-560M with AliBi (Muennighoff et al., 2022; Le Scao et al., 2023; Press et al., 2021), Pythia-410M with RoPe (Biderman et al., 2023; Su et al., 2024), and OPT-350M with learned positional embedding (Zhang et al., 2022a). All models were initialized with their pre-trained weights and fine-tuned on task sequences of length 10 to 20 bits. They were tested on sequences of up to 60 bits. Refer to Section A for experiment setup information.

Figure 2 illustrates the length generalization performance of fine-tuned BLOOMZ models using both standard and interleaved scratchpad formats, using training sequence lengths indicated by the shaded region. While the standard scratchpad method exhibits minimal improvement over not using a scratchpad, the interleaved version demonstrates perfect generalization. Notably, the sole difference between the two formats lies in the placement of the tokens in the context. The standard scratchpad format requires the model to perform index-based addressing to fetch the value of the current active bit, while the interleaved format eliminates this requirement. Section B.1 shows similar results for other models.

The observation above supports the hypothesis that the models' inability to learn arithmetic tasks stems from their failure to accurately perform index-based addressing of the input bits. In contrast, content-based addressing is inherently natural for Transformers through the attention mechanism and natural language pre-training. Next, we will further reinforce this hypothesis by introducing another modification to the standard scratchpad.

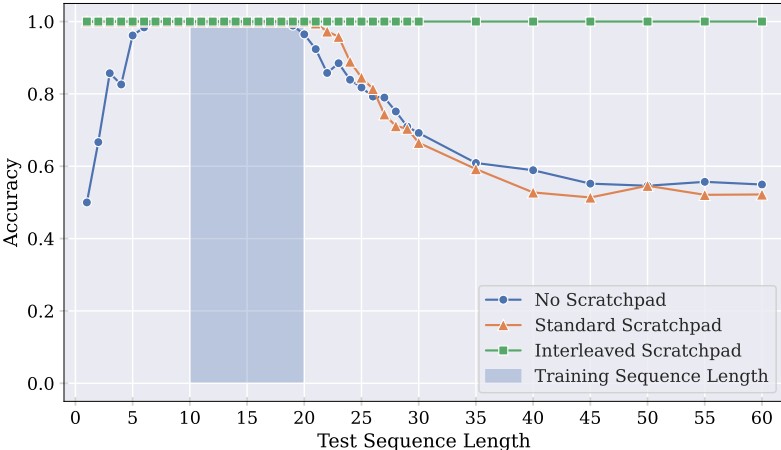

Figure 2: Length generalization performance of fine-tuned BLOOMZ-560M models on sequences of length 10 to 20 bits, using standard and interleaved scratchpad formats, as well as without a scratchpad.

## 3.2 Mnemonics

We can leverage content-based addressing in Transformers to indirectly perform index-based addressing, by adding matching "anchor" tokens before every pair of corresponding tokens in the standard scratchpad format. As they allow a model to revisit earlier information in the context window, we shall refer to these as *mnemonics*. Similar approaches are discussed in Bueno et al. (2022), Qian et al. (2022) and Zhou et al. (2023).

During training and inference, for each example of length $n$, we first randomly sample $n$ tokens from a pool of mnemonic tokens[1], then add the mnemonics before each bit in the input sequence and the running parity bits:

---

**Mnemonics**

>>> $M_1$ 1 $M_2$ 0 $M_3$ 1 $M_4$ 0 $M_5$ 0 $M_6$ 1 === $M_1$ 1 $M_2$ 1 $M_3$ 0 $M_4$ 0 $M_5$ 0 $M_6$ 1

**Mnemonics (Environment Forced)**

>>> $M_1$ 1 $M_2$ 0 $M_3$ 1 $M_4$ 0 $M_5$ 0 $M_6$ 1 === $M_1$ 1 $M_2$ 1 $M_3$ 0 $M_4$ 0 $M_5$ 0 $M_6$ 1

- - - - - - - - - - - - - - - - - - - - - - - - - - - - - - - - - - - - - - -

**Note:** *Mnemonic tokens $M_1$, $M_2$, $\cdots$ are randomly sampled without replacement from the mnemonics pool, for every problem instance.*

---

Note that in the non-environment-forced version, the model is trained to first place the matching mnemonics from the input sequence, and then use them to address the active bit at each step. Conversely, in the environment-forced version, at each step, we first append the matching mnemonic from the input sequence to the context, after which the model predicts the running parity.

Figure 3 compares the length generalization performance of fine-tuned BLOOMZ models with and without using mnemonics in the scratchpad. The results illustrate that adding mnemonics enables the model to learn the correct algorithm for solving the task, leading to perfect length generalization for sequences of up to 60 bits, while being trained on sequences of only 10 to 20 bits. Additionally, Appendix Section B.3 investigates the in-context learning performance of the parity task using mnemonics.

These results suggest that equipping a model with effective index-based addressing could be a key to enabling it to learn correct arithmetic algorithms. Interestingly, the performance

---

[1]We used all space-preceded tokens containing only English characters for the mnemonics pool.

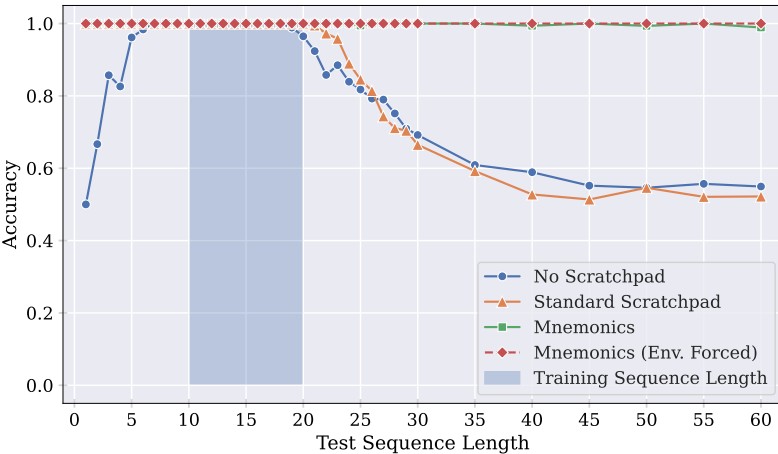

Figure 3: Length generalization performance of fine-tuned BLOOMZ-560M models with and without using mnemonics in the scratchpad.

of the model using non-environment-forced mnemonics is nearly identical to that of the environment-forced version, indicating the model's capability to both place and utilize mnemonics for indexing effectively. Similar results are reported in Appendix Section B.1 for other models. Additionally, we explore the effects of varying the interval between mnemonic tokens in Section B.2.

Using these scratchpad strategies, we also trained the same model initialized randomly instead of pre-trained on natural language. The results are shown in Figure 4. Notably, when training from random initialization, mnemonic scratchpads are ineffective. This could be attributed to the fact that successful utilization of mnemonics requires the model to perform both, *global* addressing of the relevant mnemonic, followed by *local* addressing of adjacent tokens. The latter may be an ability that persists in the length generalization setting only due to pre-training on natural language.

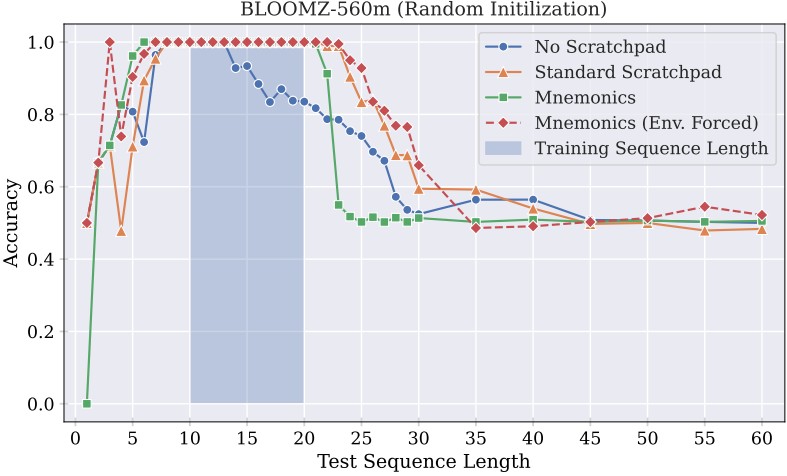

Figure 4: BLOOMZ-560M models trained from random initialization on the parity task using twice the number of epochs.

## 3.3 Analysis of attention patterns

To further analyze how the model's attention changes with and without mnemonics, we present input attribution visualizations in Figure 5, using the gradient×input method

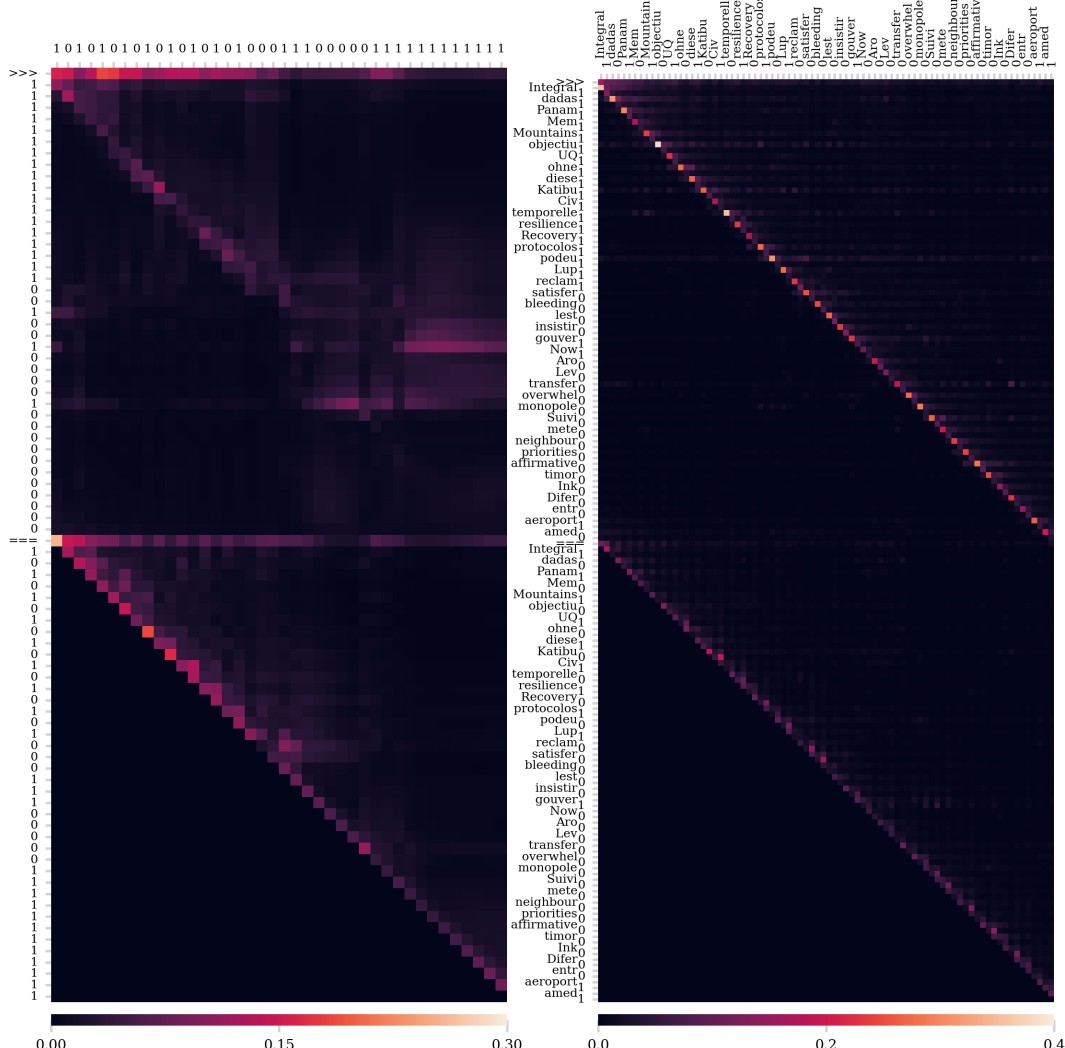

Figure 5: Input attribution visualized through the gradient×input method during performing the parity task. Models were trained on sequences of 10 to 20 bits while predicting the parity of a 40-bit sequence, shown with (right) and without (left) mnemonics. Columns represent output tokens (after ===) and rows represent all tokens in the context window. Observe the scrambled attention pattern in the left figure, after the 20th output.

(Shrikumar et al., 2016). These visualizations show aggregate attention maps, with columns representing output tokens (after ===) and rows showing all tokens in the context window. Since the model's task is to produce the running parity of the input sequence, at step $i$, it only needs to attend to the current bit (bit $i$ of the input) and the previous running parity (the last bit generated). Thus, the ideal attention map would show two diagonal lines, corresponding to these two relevant tokens. The attention maps are calculated on a sequence of length 40 for a model trained on sequences of length 10 to 20 bits.

As shown in Figure 5 on the left, immediately following the 20th bit (in-distribution length), the model fails to attend to the current bit when calculating the parity. In other words, the model has not learned a length generalizable method for indexing the correct bit at each step, thus failing at indexing outside of its training regime. In contrast, as seen in the right plot of Figure 5, when mnemonic bits are added, a near-perfect attention map is observed beyond the training regime.

## 3.4 Mnemonics variations

Finally, we study several variations of the introduced mnemonic tokens, which further support our hypothesis, as discussed below:

```
Numeric      >>> 1 b 2 a 3 b 4 a 5 a 6 b === 1 b 2 b 3 a 4 a 5 a 6 b
Constant     >>> # 1 # 0 # 1 # 0 # 0 # 1 === # 1 # 1 # 0 # 0 # 0 # 1
Non-aligned  >>> M₁ 1 M₂ 0 M₃ 1 M₄ 0 M₅ 0 M₆ 1 === M₇ 1 M₈ 1 M₉ 0 M₁₀ 0 M₁₁ 0 M₁₂ 1
Cyclic       >>> red 1 green 0 yellow 1 red 0 green 0 yellow 1
             === red 1 green 1 yellow 0 red 0 green 0 yellow 1
```

**Numeric Mnemonics:**   We use consecutive numeric indices $(1, 2, 3, \cdots)$ as mnemonic tokens for all samples. To avoid confusion between mnemonics and binary values in the sequence, we use a, b instead of 0, 1 to represent the bits. Note that this form of mnemonics corresponds to absolute positional encoding.

**Constant Mnemonics:**   A single fixed character (#) is used as the mnemonic token for all samples, during training and testing. This approach allows us to test whether the effectiveness of mnemonics is related to the attention sink phenomenon (Xiao et al., 2023), or if the model uses the mnemonic tokens as "placeholders" allowing it to store intermediate calculations in their activations.

**Non-aligned Mnemonics:**   This variant is similar to the original mnemonics with the difference that the random tokens used in the input and output do not match. Specifically, for a sequence of size $n$ bits, we sample $2n$ tokens to serve as mnemonics. We use this variant to test whether the impact of mnemonics results from making each digit unique for the model, rather than acting as positional anchors.

**Cyclic Mnemonics:**   Here, we cycle through a predetermined array of mnemonic tokens, fixed across all samples in training and testing. Specifically, we used 10 color names as mnemonics in our experiment.

Figure 6 shows the failure of the aforementioned mnemonic variants at length generalization. Note that in the environment-forced versions, all mnemonic tokens are placed in the context window of the model externally. Compared to the original randomly sampled aligned mnemonics, each variation corrupts the mnemonics' utility as positional anchors.

In the numeric mnemonics variant, the model is exposed to mnemonic tokens $1, 2, \cdots, 20$ during training, while at test time, it encounters unseen mnemonics $21, 22, \cdots$. We further explore the impact of unseen mnemonics at test time in Appendix Section B.4. Additionally, the fixed nature of numeric mnemonics across training examples may hinder length gener-

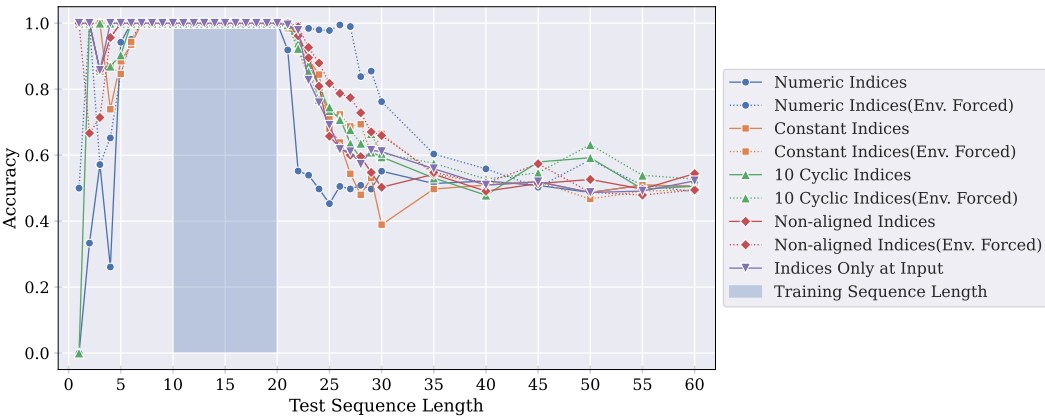

Figure 6: Length generalization performance of fine-tuned BLOOMZ-560M models on the parity task, trained on sequences of length 10 to 20 bits, using different variants of mnemonics.

alization: in contrast to the original mnemonic scheme, which randomly selects mnemonics from a large pool of tokens for each training instance, the numeric variant uses the same mnemonics for all training samples.

In the constant and non-aligned variant, anchor-based alignment between the sequence and scratchpad is eliminated entirely. Finally, cyclic mnemonics are repeating and thereby create ambiguities regarding the correct next bit to read.

Overall, these results further support our hypothesis that Transformers struggle with performing random token accesses, and demonstrate how random mnemonics can mitigate this by facilitating random access through content-based addressing of the relevant mnemonic.

## 4 Solving the Multi-digit Addition Task

This section extends our results to another arithmetic task: multi-digit addition. This task has been explored extensively in the literature with different scratchpad formats (Qian et al., 2022; Nye et al., 2021; Kazemnejad et al., 2024; Zhou et al., 2024; Xiao & Liu, 2023; Zhou et al., 2022), among others. We focus on the length generalization performance of the addition task with mnemonics in three different formats.

In our format, the addition result is initially presented in reverse order, from the least to the most significant digits. Following the symbols ###, the model then reverses this to produce the final addition result. It is important to mention that every single digit is converted to an individual token. We fine-tuned the BLOOMZ-560M model on the addition task using the specified format, training on operands with 5 to 10 digits and testing on operands with up to 14 digits.

We use the same mnemonics for corresponding digits in both operands, as demonstrated below:

```
Digit-aligned Mnemonics

No Mnemonics    >>> 1 2 + 9 === 1 2 0 ### 0 2 1
Mnemonics       >>> M₁ 1 M₂ 2 M₃ + M₂ 9 M₃ === M₂ 1 M₁ 2 M₃ 0 ### M₃ 0 M₁ 2 M₂ 1
Env. Forced     >>> M₁ 1 M₂ 2 M₃ + M₂ 9 M₃ === M₂ 1 M₁ 2 M₃ 0 ### M₃ 0 M₁ 2 M₂ 1
```

In another format, we first zero-pad the operands to ensure they have the same number of digits, then insert digit-aligned mnemonics:

```
Digit-aligned Mnemonics + Zero Padding

No Mnemonics >>> 1 2 + 0 9 === 1 2 0 ### 0 2 1
Mnemonics    >>> M₁ 1 M₂ 2 M₃ + M₁ 0 M₂ 9 M₃ === M₂ 1 M₁ 2 M₃ 0 ### M₃ 0 M₁ 2 M₂ 1
Env. Forced  >>> M₁ 1 M₂ 2 M₃ + M₁ 0 M₂ 9 M₃ === M₂ 1 M₁ 2 M₃ 0 ### M₃ 0 M₁ 2 M₂ 1
```

Lastly, we explore a format in which the mnemonics for corresponding digits of the two operands are not identical, as depicted below:

```
Non-aligned Mnemonics

No Mnemonics    >>> 1 2 + 9 === 1 2 0 ### 0 2 1
Mnemonics       >>> M₁ 1 M₂ 2 + M₃ 9 === M₃ M₂ 1 M₁ 2 0 ### 0 M₁ 2 M₃ M₂ 1
Env. Forced     >>> M₁ 1 M₂ 2 + M₃ 9 === M₃ M₂ 1 M₁ 2 0 ### 0 M₁ 2 M₃ M₂ 1
```

The length generalization performance of the addition task, both with and without the specified mnemonic formats, is shown in Figure 7. As expected, aligned mnemonics guide the model in selecting the correct digits for addition at each step. Furthermore, zero-padding simplifies the task's format by ensuring an equal number of mnemonics and digits in both operands. Overall, our findings show that similar to the simpler case of binary parity, by utilizing content-based addressing to enable index-based addressing via mnemonics, Transformer models can successfully learn the correct algorithm for the addition task.

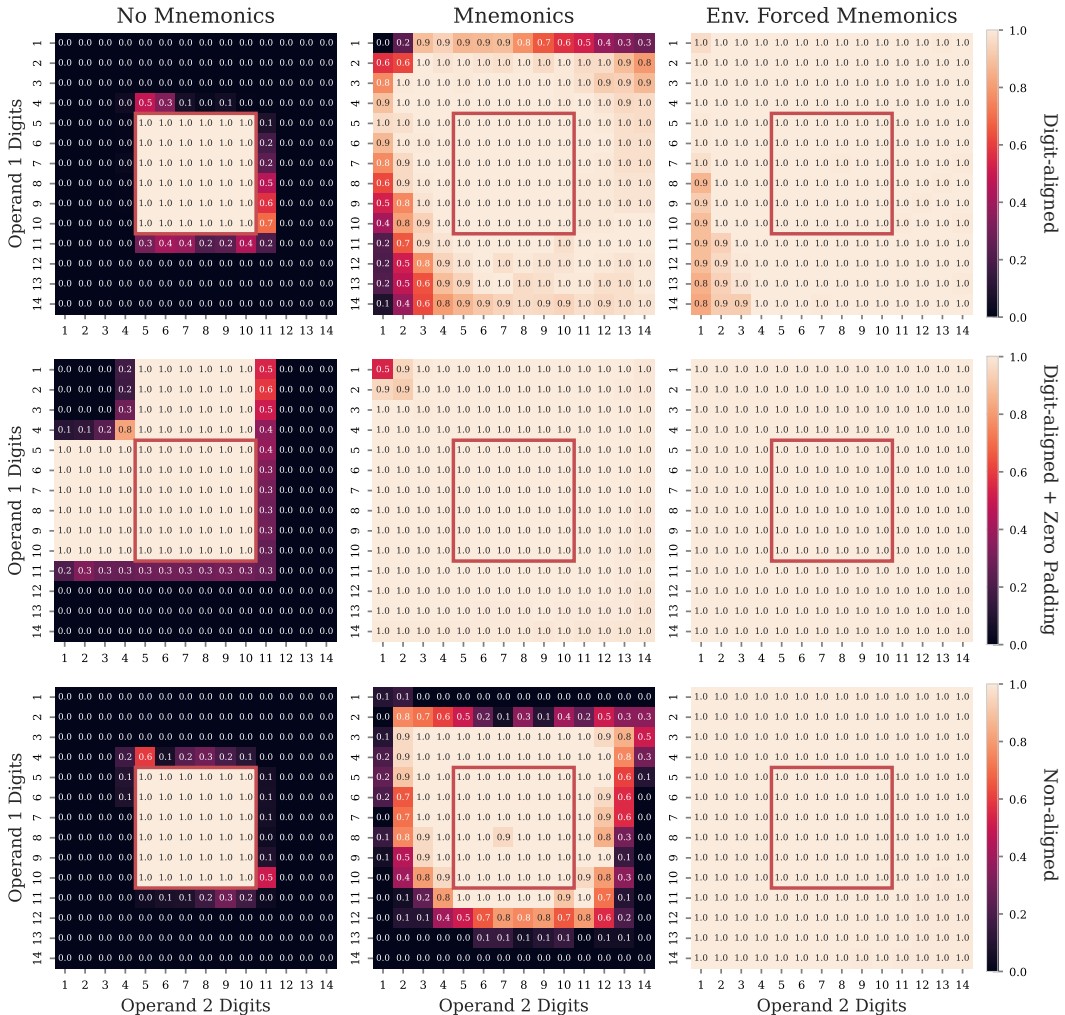

Figure 7: Accuracy of the addition task tested on operands with up to 14 digits, with models trained and evaluated with and without digit-aligned, zero-padded, and non-aligned mnemonic formats. The red box indicates the number of digits used during training.

## 5 Conclusions

We argue that, while the attention mechanism of Transformers is well-suited to perform content-based addressing into the context window, it struggles with random token accesses—a crucial capability in virtually all algorithmic reasoning tasks. We present supporting evidence for this hypothesis by demonstrating the effectiveness of methodologies that either circumvent the need for indexing, such as the interleaved scratchpad, or enable indirect random token access through content-based addressing via mnemonics. Additionally, we illustrate where and how failures in index-based retrieval manifest using attention map visualizations.

Our work demonstrates that Transformers can in fact learn to length-generalize in algorithmic tasks, such as parity and addition, as long as they are able to perform random memory access. This suggests that equipping these models with the ability to perform such index-based addressing—either into their own context window, or into an external memory—may be key to enabling them to learn algorithmic tasks more generally.

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

## A  Details of Experiments

We initialized with the pre-trained weights for training, except when specified otherwise. We used a learning rate of $1e-6$ for parity and $2e-6$ for addition, with a 1000-step warm-up. The training consists of 4 epochs, each containing 8000 training steps, with batch sizes of 64 for parity and 32 for addition tasks. We ensured an equal number of training examples for each problem length, reserving 200 samples for parity and 32 for addition from each length for evaluation. When training from random initialization, we used 8 epochs, twice the number of epochs used in our fine-tuning settings.

During training, the loss is calculated only for the target tokens (indicated by bold blue tokens in the main text). During inference, when the next token is a target, we perform greedy decoding from the model; otherwise, we place the correct token into the context window.

## B  Additional Experiment Results

### B.1  Additional models trained on the parity task

Here, we present results similar to those shown as in Figure 3 for Pythia-410M with RoPe, and OPT-350M with learned positional embeddings.

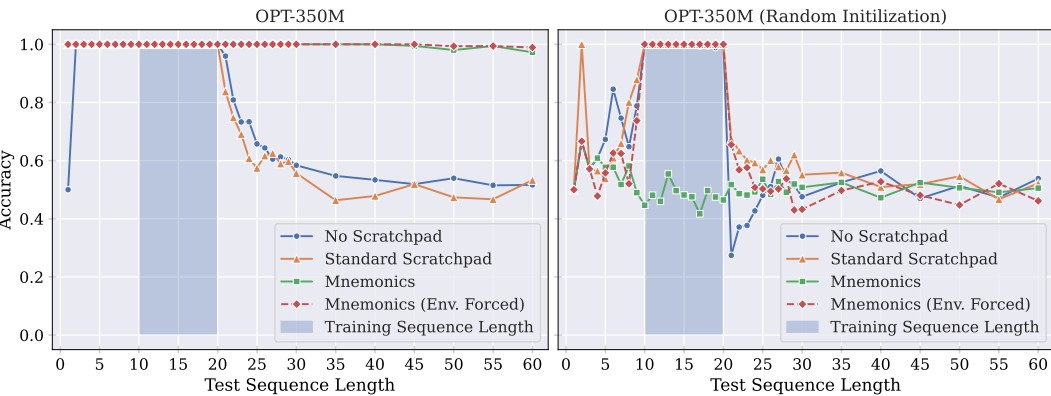

Figure 8: Length generalization performance of the OPT-350M model on the parity task using different scratchpad strategies. Left: fine-tuning; Right: training from scratch.

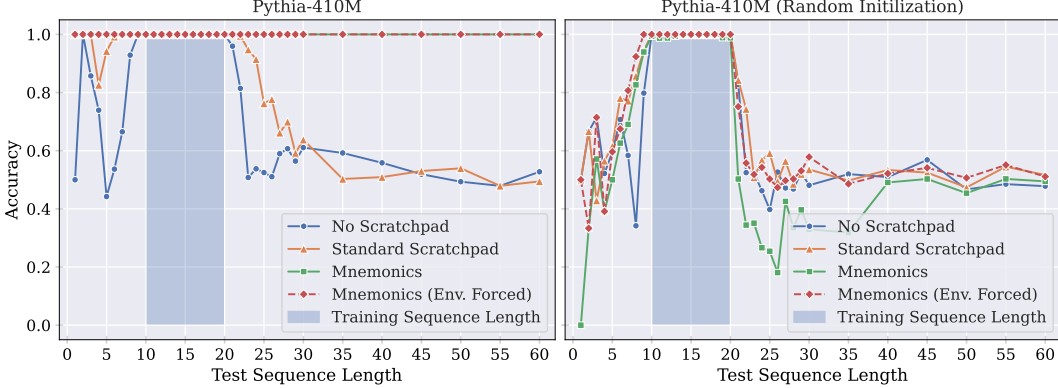

Figure 9: Length generalization performance of the Pythia-410M model on the parity task using different scratchpad strategies. Left: fine-tuning; Right: training from scratch.

## B.2  Exploring mnemonic intervals

Here, we investigate the effectiveness of reducing the number of mnemonics within the parity scratchpad. At a mnemonic interval of $i$, mnemonic tokens are inserted before every $i$-th bit in the input and output sequences. Therefore, a mnemonic interval of 1 token corresponds to the original mnemonic format described in the main text. For instance, with a mnemonic interval of 2, the format would be as follows:

---

**Mnemonics with interval of 2**
>>> $M_1$ 1 0 $M_2$ 1 0 $M_3$ 0 1 $M_4$ 0 0 === $M_1$ 1 1 $M_2$ 0 0 $M_3$ 0 1 $M_4$ 1 1

---

As shown in Figure 10, length generalization performance remains largely unaffected with mnemonic intervals of up to 3 tokens. However, when the interval exceeds 5 tokens, the impact of mnemonics begins to diminish.

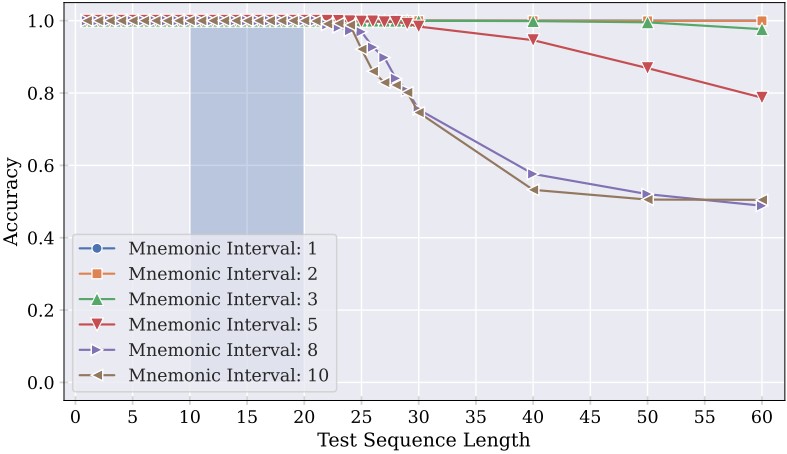

Figure 10:  Length generalization performance of fine-tuned BLOOMZ-560M models with non-environment-forced mnemonics of different intervals in the scratchpad.

## B.3  In-context learning with mnemonics

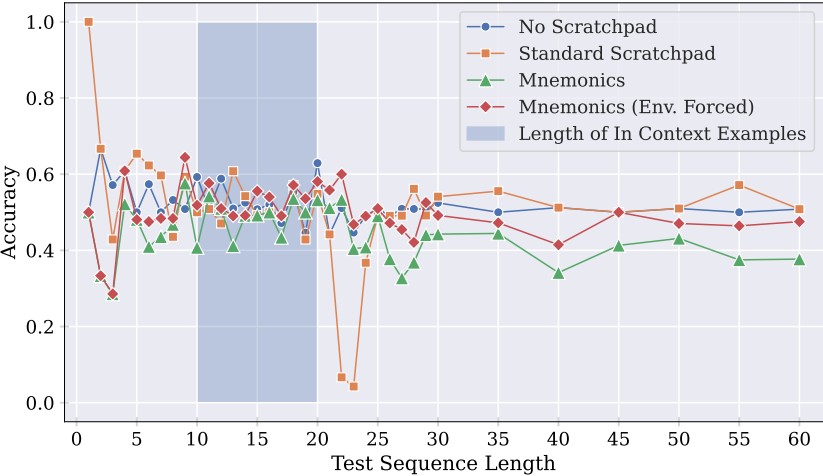

Figure 11:  Length generalization performance of a Llama2-7B model on the parity task, with in-context examples (3 examples per length) with and without mnemonics.

We investigate the in-context learning capabilities, without fine-tuning, of a larger Transformer model, Llama2-7B (Touvron et al., 2023), in performing the parity task with and without mnemonics. We use examples of lengths 10 to 20, with three examples for each length. Additionally, we preface the examples with the problem statement prompt: "Calculate the running parity of the sequence after ===". Figure 11 illustrates the model's performance with and without the use of mnemonics (refer to Section 3.2). Similar results were also observed with Llama2-7B-chat and BLOOMZ-7.1B models.

## B.4 Unseen (OOD) mnemonics at test time

In this section, we investigate whether the model treats mnemonics merely as positional anchors, disregarding their values, or if it learns to memorize the mnemonic tokens for indexing. Following the methodology described in Section 3.2, we fine-tune a model using single-token English words as mnemonics. In contrast, at test time, we use single-token integers as mnemonics.

Figure 12 presents the results of length generalization performance for models evaluated on in-distribution and out-of-distribution mnemonics. It shows that performance degrades when a model is evaluated on unseen, semantically novel mnemonics. This suggests that the learned approach to using mnemonics still relies on token values.

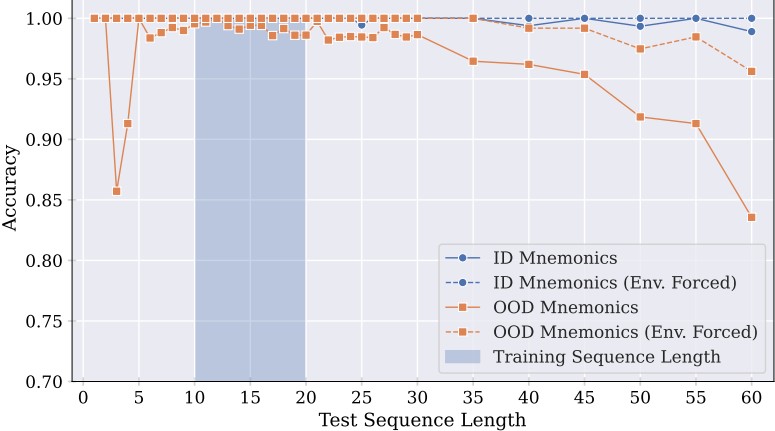

Figure 12: Length generalization performance of fine-tuned BLOOMZ-560M models, tested using in-distribution (ID) and out-of-distribution (OOD) mnemonics. Note that the y-axis is truncated, with values ranging from 0.7 to 1.

