# OpenReview forum: "Your Context Is Not an Array: Unveiling Random Access Limitations in Transformers"
_colmweb.org/COLM/2024/Conference — COLM_

### Official Review · Reviewer_VRhj · 2024-05-09

**Rating:** 8
**Confidence:** 4
**Ethics Flag:** 1

**Summary:**

This paper studies length generalization of transformers using parity and addition as prototypical tasks and identifies _index-based access_ as a cause of failures of generalization.  In particular, they show that a scratchpad does not help with parity, but that an _interleaved_ scratchpad enables near-perfect length generalization.  To fully test the index-based access idea, they introduce "mnemonic" tokens into the input and scratchpad, which provide a direct anchor between output tokens and the corresponding input token.  They show that this method also enables perfect generalization (NB: when the mnemonic token pool is large enough and they are randomly sampled without replacement each time; see questions below).  Other forms of mnemonic do not work, suggesting that it really is the model's ability to know which position to attend to that hinders length generalization in the normal case (an attention pattern analysis also supports this).  Similar results are shown for addition as well.  These results are a clever use of scratchpad format to do detailed analysis of the factors which prevent and enable length generalization in transformers; while the authors do not explicitly recommend these formats for specific tasks, they are to be commended for the careful and detailed analysis.

**Questions To Authors:**

* I really enjoyed the various mnemonic ideas, but was puzzled by this: "For example, with the numeric indices, the model is exposed to mnemonic tokens 1, 2, · · · , 20, but encounters unseen mnemonics 21, 22, · · · in test time."  How is this different for numerical mnemonics than for the mnemonics results reported in Figure 3?  I think I eventually figured this out: the $M_i$ in Fig 3 are sampled from the same overall pool for each instance, so it is possible that every $M_i$ will be seen during training even if no inputs of a certain length have been seen.  I think some wording in 3.1 could help clarify this, or at least-referencing the **Note:** in the pnemonics figure when discussing the numerical indices later on (it at least took me awhile to unpack all of this).  Also: I guess you wouldn't be able to length generalize beyond the _total_ number of mnemonics for the same reason.  E.g. if you have 100 total $M_i$ tokens and train on seqs of length up to 20, you'd expect length generalization up to 100 but then not beyond; is that right?

* Numeric mnemonics: it strikes me as interesting and a little perplexing that the models do not generalize to longer lengths with numerical index mnemonics.  (If the model learned a general rule about "look to the right of the same token" [or the equivalent generation rule in the non-environment-forced case], it shouldn't matter which tokens it has and hasn't seen as mnemonics.)  Do you know if all of the digits you use are full tokens in the relevant vocabs, or is it possible that some longer digits are getting split up?  Related to this and my above question: did you try an "O.O.D. mnemonic generalization", where the mnemonic tokens at inference time can be different, but are the same ones as in the first experiment (i.e. single tokens with whitespace first)?

**Reasons To Accept:**

* Detailed analysis of failure and success modes of language models on length generalization parity and addition.
* Identifies random/position-based access (as opposed to content-based access) as a difficulty.
* Construction of careful scratchpad formats using "mnemonic tokens" both enables length generalization and helps to diagnose failure modes.

**Reasons To Reject:**

* The introduction of position- vs content-based addressing was a bit fast and loose (e.g. the discussion of a PP-attachment ambiguity at the bottom of page 1).
* Only reports results of fine-tuning; I'd be curious to see if their methods also work in an in-context learning setup.

---

> ### Author Rebuttal · Authors · 2024-05-30
>
> > The introduction of position- vs content-based addressing was a bit fast and loose (e.g. the discussion of a PP-attachment ambiguity at the bottom of page 1).
>
> We will add more clarification on positional vs content-based addressing in natural language tasks. This is also related to [this reference](https://arxiv.org/abs/2104.06644) suggested by Reviewer 1, which we will include in the final version.
>
> >Only reports results of fine-tuning; I'd be curious to see if their methods also work in an in-context learning setup.
>
> The models in the size range we explored do not perform well with in-context learning on these tasks (even though they do very well when trained with mnemonics).
> We are exploring the possibility of in-context learning for mnemonics using larger models.
>
> > is it possible that some longer digits are getting split up?
>
> We can confirm that this is not the case for numeric mnemonics. We specifically ensured that all mnemonics presented in the paper were encoded as single tokens.
>
> > did you try an "O.O.D. mnemonic generalization"
>
> Thank you for the interesting suggestion. Based on your feedback and also comments from Reviewer 1, we will include discussions and results for scenarios where unseen mnemonic tokens are introduced during test time. This will assess whether the model has learned to disregard the specific values of the indices and treat them as mere positional anchors. We also refer you to our response to Reviewer 1 for a related discussion.
>
> >I think some wording in 3.1 could help clarify this
>
> Following your suggestion, we will include clarifications on the OOD nature of numerical mnemonics.

---

> > ### Comment · Reviewer_VRhj · 2024-06-05
> > **Thanks for the reply!**
> >
> > I thank the authors for the detailed and helpful reply, and enjoyed reading the related discussion with another reviewer on O.O.D. mnemonics.  My score is already very high, indicating my belief that this paper should definitely be accepted, and so I will leave it as is.

---

### Official Review · Reviewer_B5h5 · 2024-05-10

**Rating:** 7
**Confidence:** 3
**Ethics Flag:** 1

**Summary:**

In this paper, the authors study transformer's ability to perform length generalisation when the task to be solved involves index-based search.

To do so, the authors first focus on the binary parity task and fine-tune three models (which use three different positional encoding mechanisms: AliBi, RoPE, and learned positional embeddings). The three models are able to perform the task almost perfectly for lengths similar to ones seen during training but completely fail when increasing the length.

The authors then simplify the task by allowing the models to keep a scratchpad. However, as this still involves performing index-based addressing, the models still fail. But when having an interleaved scratchpad, as the model only needs to attend to the two previous tokens, at each step, its accuracy its almost perfect.

To further illustrate this, the authors also experiment inserting mnemonics in the sequence (adding matching random anchor tokens before pairs of corresponding tokens) so that the model can solve the task by performing content-based addressing. The transformer models are able to solve this task with very good accuracy, showing that the attention mechanism is well suited for content-based addressing but not for index-based addressing.

The authors also show through input attribution plots, that the attention mechanism clearly fails to attend to the correct bits when the length of the sequence is bigger than the ones seen during training.

To conclude the authors, perform similar experiments on a different task: multi-digit addition, which shows similar results.

**Reasons To Accept:**

- The paper is very well written and is easy to follow.
- The authors study an interesting flaw of transformers (or current positional encoding mechanisms) that might prevent their ability to perform algorithmic tasks.
- The authors present interesting evidence that show that transformer models (with the tested positional encoding mechanisms) are very good at performing content-based addressing but fail to length generalise when having to perform index-based addressing.
- The paper provides in-depth analysis of the experiments.

**Reasons To Reject:**

- I think the paper could be even more interesting if the authors explored possible solutions to improve trasnformer's random token access ability or at least proposed some possible solutions for future work.
- The paper could also explore "real-world" tasks for which transformers might struggle due to the lack of index-based search capabilities.

---

> ### Author Rebuttal · Authors · 2024-05-30
>
> > I think the paper could be even more interesting if the authors explored possible solutions to improve trasnformer's random token access ability or at least proposed some possible solutions for future work.
>
> Thank you for your suggestion.
> Our work identifies random-accessing as a fundamental issue that currently lacks a general solution. A more principled approach, although out of scope for this work, would be a learned procedural search capability. We will also include this as a direction for future research.
>
> > The paper could also explore "real-world" tasks for which transformers might struggle due to the lack of index-based search capabilities.
>
> While the purpose of this work is to identify the core issue, we acknowledge that solving random memory access could benefit many real-world tasks, including most algorithms. We will include a discussion on this topic in the paper. As previously mentioned, we consider learning to procedurally search in memory as an important direction for future research.

---

> > ### Comment · Reviewer_B5h5 · 2024-06-04
> >
> > Thanks for the reply.
> >
> > Having read the authors' rebuttal, I've decided to maintain my score.

---

### Official Review · Reviewer_yGJY · 2024-05-10

**Rating:** 7
**Confidence:** 4
**Ethics Flag:** 1

**Summary:**

This paper argues that certain reasoning tasks, even with a scratchpad/chain-of-thought, are difficult because they depend on precise position information in a way that natural language does not. It studies two example tasks, parity and (more briefly) addition, and explores several variations on these tasks to see what helps and what doesn't.

The three things that help are:

- For parity, forcing the model to copy the input and fine-tuning it to generate the state interleaved with the copied input.
- For both tasks, prompting it with the input interleaved with "mnemonics" and fine-tuning it to generate the state interleaved with the mnemonics.
- Same, but forcing the second generation of the mnemonics.

Many contrastive experiments are performed to show that it is accessing information at a precise, non-local position that is difficult.

**Questions To Authors:**

The paper seems to be using the wrong font.

For parity, why didn't you try the interleaved scratchpad without environment-forcing?

Although you tried several different established position embeddings, why not encode the "mnemonics" into the position embedding instead of interleaving them? That is, for your parity example, you could prompt the model with word/positions:

1/1 0/2 1/3 0/4 0/5 1/6 1/7 ===/8

and then the completion could be

1/1' 1/2' 0/3' 0/4' 0/5' 1/6' 0/7'

where 1' is the same position embedding as 1, but possibly modified somehow.

**Reasons To Accept:**

This is an interesting phenomenon and the experiments are fairly convincing.

**Reasons To Reject:**

The experiments are limited to two tasks, and the addition experiments are not as thorough as the parity experiments.

---

> ### Author Rebuttal · Authors · 2024-05-30
>
> > The paper seems to be using the wrong font.
>
> We thank the reviewer for their attention to detail. The incorrect font usage resulted from an issue with importing a package, which we have now corrected.
>
>
> > For parity, why didn't you try the interleaved scratchpad without environment-forcing?
>
> While mnemonic tokens can optionally be environment-forced, in the interleaved scratchpad version, the input sequence bits are always environment-forced, as without the model would not know the problem input. We hope this clarifies the reviewer's question.
>
>
> > Although you tried several different established position embeddings, why not encode the "mnemonics" into the position embedding instead of interleaving them? That is, for your parity example, you could prompt the model with word/positions:
> 1/1 0/2 1/3 0/4 0/5 1/6 1/7 ===/8
> and then the completion could be
> 1/1' 1/2' 0/3' 0/4' 0/5' 1/6' 0/7'
> where 1' is the same position embedding as 1, but possibly modified somehow.
>
> Thank you for the suggestion. If you are referring to an embedding-level mnemonic, such as appending real-valued vectors to positional embeddings, it's an interesting idea. We believe this could potentially help with length generalization, similar to the effects observed with token-level mnemonics.
>
> However, if you mean adding "/position" as token-level mnemonics to prompt the model with 'word/positions,' this approach aligns with the numeric mnemonic format discussed in Section 3.3.
>
> If we have misunderstood your suggestion, please let us know so we can address it further.

---

### Official Review · Reviewer_Uz62 · 2024-05-14

**Rating:** 7
**Confidence:** 3
**Ethics Flag:** 1

**Summary:**

The paper is focused around the issue of length generalization in LLM models in the arithmetics tasks. The authors’ hypothesis is that this problem is strongly connected to the ‘random’ addressing problem, i.e. the ability to exactly address a token by its position. Unlike for natural language, for arithmetic tasks the positional information is equally important to the token content; hence this is less of a problem for the NL tasks than for the arithmetics.

As a motivating example, the paper starts with a task of calculating XOR of a bit sequence. Arbitrary input-length calculation is not solvable by a Transformer as it would have a fixed computational power; yet, by adding the chain-of-thought (scratchpad) output with the running partial answer and an incoming bit, that should be alleviated. However, the experiments show this not to be the case. In contrast, in the case where the incoming bit is externally fed, in the same format, the model works perfectly across lengths. From that, the authors conclude that the main issue is the inability to retrieve the correct input bit.

In the following experiments, the authors improve the addressing capability by adding special mnemonic tokens in the sequence.. Those are randomly sampled (w/o replacement) tokens that serve as addressing anchors. This modification seems to generalize well across sequences even when the mnemonic tokens are not externally forced. They also work better than numeric, constant, non-aligned, and cyclic anchors.

In the next steps the authors also provide attention pattern analysis, experiment with the addition tasks. Appendix gives bits of additional insights, e.g. experiments with the mnemonic-encoded intervals.

**Questions To Authors:**

* Zooming in on the ablation where ‘mnemonics’ anchors are replaced by the numerics (which are akin to absolute positional embeddings). Now, there are two differences between those two: (a) randomized order of the mnemonics vs strict order sequential of numerics, and (b) ‘out-of-vocabulary’-ness of the numerics on longer sequences (e.g. 22 is not seen at training).

Now, can we nail down which one makes the mnemonics work? For instance, the randomized order can make the model be really careful to pay attention to the input, while with monotone numbers it can just put the a priori known order w/o looking at the input even. Alternative explanation is that the model indeed is lost when it gets an OOD anchor in the input.

Can we design an experiment where we differentiate the two above cases? Eg, always use monotone numerical anchors from the range of 0…100, but starting at a random position (10, 11, 12, …, 30 for one sequence and 0, 1, 2, …, 20 for another). This way the OOD issue disappears (for sequence of length 30 we can go w/ 0…29) but the monotonicity is preserved.

(Maybe this exact proposal is wrong, but I am interested if we can come up w/ a way to distinguish the causes for the behavior).

* The paper says that “.. the introduction of mnemonics is not proposed as a practical fix, it highlights the underlying issue and reinforces our hypothesis.” Could you please expand (a) why you would deem it impractical, and (b) what could be a practical step from here?

* The paper deals with two arithmetics-based experiments where the alignment is very much fixed. Can we carry over any ideas from here to tasks with less strict alignment? Or even would it be any useful beyond arithmetics, for eg NMT tasks?

* I assume that the interleaved scratchpad has forced input bits, is that correct?

* Suggestion: Please consider using different labels for points in the figures, not solely relying on the color for differentiating them.
* Comment: the introduction has a bit of a discussion on relative importance of words vs their positions. I feel that [1] would be a fitting citation to show that one can go a very long way in natural language w/o even having correct positional information.

 [1] https://arxiv.org/abs/2104.06644

**Reasons To Accept:**

* I really like that the paper provides a focused, well-explained and in-depth study of a single problem. It tries to motivate the problem, nail down the root cause and find a grounded solution for it.
* To my taste, the experiments are clever, illuminating and nicely support the hypothesis/conclusions.
* The paper allows us to get a better inner model of Transformers

**Reasons To Reject:**

Playing a devil’s advocate:
* What I think is missing is a motivation why/when this is important beyond arithmetics. If that is only a problem for the arithmetics, this can be, in principle, solved by tool-using a calculator/python interpreter.
* Authors highlight that their proposed mnemonic solution is not a practical one. What does those findings tell us about a potential practical solution?

---

> ### Author Rebuttal · Authors · 2024-05-30
>
> Thank you for your constructive feedback and suggestions.
>
> >What I think is missing ...
>
> While we study these specific tasks, the problem should affect any task that requires random memory access, which could include many NLP tasks, such as counting occurrences of references. We will clarify this point in the paper.
>
> >Authors highlight that their ...
>
> Our work highlights that this is a fundamental problem for which there is currently no general solution. It also suggests that learning to procedurally search in memory could be a fruitful direction toward a potential solution.
>
> > Zooming in on the ablation ...
>
> We acknowledge your observation and note that, besides being monotonic, numerical indices are also fixed across all samples, making them 1) fixed, 2) monotonic, and 3) partially OOD during test time.
>
> While our experiments suggest that your idea of using a randomized starting point improves length generalization, it remains unclear whether this is due to alleviating the OOD problem or disrupting fixed index patterns.
>
> To further isolate these factors, we propose using OOD tokens at test time, while using a randomized starting position for numerical indices during training. This will assess if the model has learned to disregard the specific values of the indices and treat them merely as positional anchors.
>
> We will include a discussion and corresponding experiments in the final version of the paper.
>
> > The paper says that “.. the introduction of ...
>
> We consider this strategy impractical for two reasons. It doubles the token count required for tasks, and, as seen in the scratchpad design for addition, it requires careful placement of mnemonics to align the digits in the scratchpad format. This could become tedious for more involved algorithms, such as multiplication.
>
> > The paper deals with two arithmetics-based ...
>
> As we discussed above, a more principled solution than mnemonics, although out of scope for this work, would be a learned procedural search capability, which should be applicable even in tasks without strict alignment patterns.
>
> > I assume that the interleaved ...
>
> Yes, the input bits are environment-forced, and the model predicts the running parity at each step.
>
> > Suggestion: Please consider using ...
>
> Thank you. We will apply this in the final version.
>
> > Comment: the introduction has ....
>
> The point made in the paper indeed supports our argument on content-based vs. index-based addressing, which we will include in the final paper.

---

### Decision · Program_Chairs · 2024-07-10

**Decision:**

Accept

**Comment:**

The reviewer appreciated this analysis of length generalization, and found the results interesting and useful.
The authors may find this paper on markups somewhat relevant (and possibly related papers which use different markings to "help the model count"): https://arxiv.org/abs/2208.11445